# Emerging Role of Long, Non-Coding RNA Nuclear-Enriched Abundant Transcript 1 in Stress- and Immune-Related Diseases

**DOI:** 10.3390/ijms26094413

**Published:** 2025-05-06

**Authors:** Xingliang Liu, William Haugh, Ziqiang Zhang, Jianguo Huang

**Affiliations:** 1Earle A. Chiles Research Institute, Providence Cancer Institute, Portland, OR 97213, USA; xingliang.liu@providence.org (X.L.); william.haugh@providence.org (W.H.); 2Department of Respiratory and Critical Care Medicine, Tongji Hospital, Tongji University School of Medicine, Shanghai 200065, China

**Keywords:** long, non-coding RNAs, NEAT1, stress-related diseases, immune cells, immune-related diseases, biomarker, therapeutic target

## Abstract

Long, non-coding RNAs (lncRNAs) are a class of RNAs exceeding 200 nucleotides in length, lacking the ability to be translated into proteins. Over the past few decades, an increasing number of publications have established lncRNAs as potent regulators in a broad spectrum of diseases. They modulate the expression of critical genes by affecting transcription, post-transcription, translation, and protein modification. This regulation frequently involves the interaction of lncRNAs with various molecules, such as proteins, RNA, and DNA. lncRNAs are involved in diseases where stress is a significant factor. In recent years, lncRNAs have been identified as regulators of both innate and adaptive immune responses, playing significant roles in the onset and progression of diseases. Additionally, lncRNAs hold potential as biomarkers or therapeutic targets for numerous stress- and immune-related diseases. lncRNA nuclear-enriched abundant transcript 1 (NEAT1) is a notable example. This review consolidates the latest findings about the role of lncRNA NEAT1 in stress response and immune cell function in non-cancer diseases. It summarizes studies on NEAT1 regulating stress response, both innate and adaptive immunity, and its potential as a biomarker and therapeutic target for stress- and immune-related diseases.

## 1. Introduction

Accumulating studies have begun to appreciate the role of lncRNA NEAT1 in regulating the stress response, such as the response to genotoxic stress or oxidative stress, as well as to hypoxia. Some studies further indicate that NEAT1 may serve as a biomarker or therapeutic target for stress-related diseases. In recent years, mounting evidence has demonstrated the role of lncRNA NEAT1 in immune-related diseases, in addition to cancers. Viral infections directly induce the expression of NEAT1, resulting in an immune response against the virus. Consequently, NEAT1 expression is significantly upregulated in peripheral blood mononuclear cells (PBMCs) and serums of patients infected with various viruses, including COVID-19 and human immunodeficiency virus 1 (HIV-1). Additionally, the RNA levels of NEAT1 are often positively correlated with the severity of the infection, such as with sepsis. All these studies highlight that NEAT1 may serve as a biomarker for viral infections. NEAT1 is also highly expressed in a subset of monocytes and has been used as a biomarker for these subsets in single-cell sequencing analyses. NEAT1 can play a role in both innate immunity and adaptive immunity. NEAT1 can induce the activation of the inflammasome and activate macrophages. NEAT1 can induce cluster of differentiation-4-positive (CD4^+^) T cells to differentiate into T helper 17 (Th17) cells and is also critical for regulating the T helper 1/T helper 2 (Th1/Th2) balance. Furthermore, NEAT1 has potential as a therapeutic target in immune-related diseases. Therefore, we reviewed studies dissecting the roles of NEAT1 in the stress response, innate and adaptive immune cell function, and as a biomarker and therapeutic target in stress- and immune-related diseases.

## 2. Overview of Long, Non-Coding RNAs (lncRNAs)

Long, non-coding RNAs (lncRNAs) are a group of RNAs longer than 200 nucleotides. lncRNAs lack the capability to translate into full proteins. However, a subset of lncRNAs can translate into small peptides, which show biological functions [1,2,3]. Over the last few decades, mounting evidence has demonstrated the role of lncRNAs in various cell processes, including cell proliferation, apoptosis, invasion, and metastasis [4,5,6,7]. lncRNAs can be classified as antisense, divergent, and intergenic lncRNAs, based on the direction of their transcription and their proximity to neighboring genes [8]. The number of identified lncRNAs has dramatically increased in recent decades. For instance, approximately 95,243 lncRNA genes and 323,950 lncRNA transcripts have been identified in humans from a study in 2023 [9]. Advances in new sequencing techniques and new bioinformatics tools will likely reveal even more lncRNAs [10]. The prevalence of a large amount of lncRNAs underscores their importances in biology and disease. It is also important to note that the expression pattern of lncRNAs exhibits greater tissue specificity and cell-type selectivity compared to coding genes, suggesting their roles may be cell-type-specific [11].

The cellular stress response refers to the various molecular changes that cells undergo when exposed to environmental stressors, such as oxidative stress. When cells encounter stress, they activate a range of protective mechanisms, such as the production of stress proteins, e.g., heat shock proteins (HSPs), to ensure proteins maintain their proper structure and function [12]. If the stress is too severe or prolonged, cells will undergo programmed cell death to prevent damage to the organism [13]. This process is tightly regulated and ensures that damaged cells are removed in a controlled manner. However, uncontrolled stress will lead to disease. Such as in the case of Alzheimer’s disease (AD), where oxidative stress plays a critical role in its pathogenesis [14]. Mounting evidence indicates the role of lncRNAs in the regulation of stress-related diseases. They are involved in various cellular stress responses, including oxidative stress, hypoxia, and genotoxic stress [15].

The innate and adaptive immune responses are critical for defending the body against pathogens. The innate immune response is the first line of defense and activates adaptive immune responses, leading to the elimination of pathogens. The innate immune response involves immune cells, including macrophages, neutrophils, dendritic cells, natural killer (NK) cells, mast cells, eosinophils, and basophils. The adaptive immune response involves immune cells, including B cells and T cells. Within the immune system, various types of immune cells work together to trigger an immune response. Extensive evidence has begun to appreciate the role of lncRNAs in immune cells, including their development, differentiation, activation, and effector functions. Therefore, there are new opportunities to target these critical lncRNAs to modulate immune responses and suppress immune-mediated pathogens, paving the way for innovative therapeutic strategies. Extensive reviews have summarized the role of lncRNAs in immune systems under both physiological and disease conditions [16,17,18].

In this review, we aim to provide a comprehensive overview of one lncRNA involved in the stress response and multiple immune cell types: nuclear-enriched abundant transcript 1 (NEAT1). By examining its functions in stress regulation and immune cell regulation, as well as its potential as a biomarker and therapeutic target, we can gain insights into its role in stress- and immune-related diseases.

## 3. Structure and Function of NEAT1

NEAT1 is conserved across different species and is highly expressed and specifically localized in paraspeckles. Additionally, NEAT1 is essential for the paraspeckle complex [19,20,21]. NEAT1 has two isoforms, NEAT1_1 and NEAT1_2, which are produced through alternative 3′-end processing (Figure 1). The two isoforms share a common promoter; therefore, NEAT1_1, the short isoform, completely overlaps with the 5′ end of the longer isoform, NEAT1_2. NEAT1_1 is generated through termination by a polyadenylation signal (PAS) during transcription [22]. In humans, NEAT_1 is approximately 3.7 kilobases (kbs) long, and NEAT1_2 is about 23 kbs. NEAT1_1 is more abundant than NEAT1_2 in cells. Studies have shown that the two isoforms are localized in diverse structures in the nucleus [22,23,24]. NEAT1_2 is exclusively expressed in nuclear paraspeckles. NEAT1_1 can localize to other structures, such as microspeckles [23]. Furthermore, the two isoforms have diverse functions in biology. NEAT1_2 is essential for paraspeckle assembly and can regulate RNA transcription through regulating the paraspeckles [20,25]. In this review, we will introduce several studies which demonstrate that NEAT1_2 can modulate gene expression through regulating transcription at the targeted site in the paraspeckles. In contrast, NEAT1_1 is nonessential for paraspeckle assembly. NEAT1_1 regulates the expression of the targeted genes by interacting with other proteins in the cells. For instance, It has been found that NEAT1_1 promotes sarcoma metastasis by interacting with a RNA splicing protein, KT-type splicing regulatory protein (KHSRP) [7]. There are other studies further supporting the evidence that the two isoforms play diverse roles in the cells. One study showed that the relative abundance of NEAT1_1 and NEAT1_2 differs between aggressive and non-aggressive neuroblastoma cancer cells. For instance, high-risk MYCN-amplified aggressive cancer cells express high levels of NEAT1_1 relative to NEAT1_2, while non-aggressive cancer cells with a more differentiated phenotype express relatively high levels of NEAT1_2. They further found that NEAT1_1 acts as an oncogene and NEAT1_2 as a tumor suppressor in neuroblastoma [26]. However, it is noteworthy that the unique feature of NEAT1_1 overlapping with NEAT1_2 makes it difficult to dissect the function of individual isoforms. Therefore, most studies, including those cited in this review, are focused on dissecting the functions of NEAT1 without properly modulating the expression of individual isoforms. Several strategies have been developed to modulate the expression of individual isoforms of NEAT1 for studying the function of individual isoforms in biology [23,26,27]. Therefore, it is important to carefully choose the right method for exploring which isoform of NEAT1 is important for the observed phenotype.

## 4. NEAT1 in Stress Regulation

In response to stimuli that damage intracellular molecules and their functions, cells trigger a stress response to restore homeostasis to an adaptive state or induce cell death. These stress stimuli include DNA damage by genotoxic agents, oxidative stress, and hypoxia. A figure summarized the publications showing the roles of NEAT1 in the regulation of these stresses (Figure 2).

### 4.1. NEAT1 Regulates Genotoxic Stress

Chemical agents, UV, and irradiation can result in DNA damage, thereby inducing genotoxic stress. NEAT1 regulates genotoxic stress by controlling DNA damage response (DDR) pathways, cell cycles, and cell death pathways. In response to genotoxic stress in cells, TP53, the guardian of genome stability, is activated and induces cell cycle arrest, DDR, senescence, and apoptosis [28]. When cells experience DNA damage, NEAT1 is transcriptionally upregulated by TP53 [29,30]. Upregulated NEAT1 accumulates at the sites of damage and facilitates the repair process [31]. Mamontova V et al. dissect the underlying mechanisms by which NEAT1 regulates this process [31]. In addition to the transcriptional upregulation of NEAT1, DNA damage also increases the N6-methyladenosine (N6) marks in NEAT1. This mark forces structural alterations in NEAT1, directing its accumulation at promoter-associated DNA double-strand breaks (DSBs), facilitating the repair process. Additionally, TP53 can induce the formation of NEAT1-containing paraspeckles, which help manage the replication process. The interaction between NEAT1 and TP53 creates a feedback loop that modulates the cellular response to DNA damage and oncogene activation. Therefore, the knockout of NEAT1 sensitizes cells to DNA-damage-induced cell death and impairs tumorigenesis [29,30]. Another study specifically found that the short isoform, NEAT1_1, is important for maintaining genome integrity through interacting with the Tudor Interacting Repair Regulator (TIRR)/TP53-binding protein 1 (53BP1) complex. Furthermore, they linked the connection between NEAT1 and neurodegenerative diseases, which are often affected by stress [32,33,34,35,36,37,38]. They showed that TAR DNA-binding protein 43 (TDP-43), which is involved in neurodegenerative diseases, promotes the expression of NEAT1 by modulating the TIRR/53BP1 complex.

Mohammad G et al. showed that NEAT1 can also translocate into mitochondria, where damage can lead to diabetic retinopathy [39]. They found that high glucose can increase NEAT1 expression in mitochondria. The knockdown of NEAT1 will reduce glucose-induced damage to the mitochondrial membrane and DNA, suggesting that NEAT1 protects mitochondrial homeostasis and reduces the formation of degenerative capillaries in diabetic retinopathy. Therefore, NEAT1 can affect stress-related diseases through regulating genotoxic stress.

### 4.2. NEAT1 Regulates Oxidative Stress

Oxidative stress occurs in cells when there is an imbalance between the production of reactive oxygen species (ROS) and the cell’s ability to detoxify these reactive intermediates or repair the damage. ROS, which include free radicals like superoxide anion (O_2_^−^) and hydroxyl radical (OH), as well as non-radical species like hydrogen peroxide (H_2_O_2_), can damage cellular components, such as lipids, proteins, and DNA [40].

Oxidative stress can transcriptionally regulate NEAT1 expression. For instance, H_2_O_2_ treatment can induce the expression of NEAT1 [41,42,43,44,45]. Another study showed that (-)-epigallocatechin-3-gallate (EGCG), a green tea polyphenol, can increase the expression of NEAT1 by inducing ROS-mediated oxidative stress [46]. Zhao M et al. showed that oxidative stress induced by mitochondrial ROS can impair the expression of NEAT1 in mesenchymal stem cells [47]. Dysregulated NEAT1 can then regulate the cellular response to oxidative stress in stress-related diseases. In a model of autism spectrum disorder (ASD), He C et al. showed that NEAT1 promotes valproic acid (VPA)-induced ASD in rats through upregulating apoptosis and oxidative stress [48]. They demonstrated that NEAT1 can induce the transcription of ubiquitin protein ligase E3A (UBE3A) by recruiting the transcription factor Yin Yang 1 (YY1) to its promoter, thereby inducing oxidative stress-induced neuronal damage. Mitochondria-targeting metal–organic frameworks loaded with ruthenium nanozyme (Mito-Ru MOF) can scavenge free radicals and ROS, thereby relieving oxidative stress [49]. A single Mito-Ru MOF treatment downregulates NEAT1 and inhibits the production of ROS, thereby reducing Freund’s adjuvant (CFA)-induced temporomandibular joint disorders (TMD). Guo X et al. demonstrated that the inhibition of NEAT1 can reduce oxidative stress-induced apoptosis through upregulating miR-124-3p [50]. The excessive decomposition of the extracellular matrix (ECM) in nucleus pulposus (NP) cells can result in the pathological process underlying intervertebral disc degeneration (IVDD). Shang L et al. showed that docosahexaenoic acid (DHA) can reduce the excessive degradation of the ECM in NP cells in response to oxidative stress by decreasing the expression of NEAT1. The inhibition of NEAT1 reduced H_2_O_2_-induced oxidative injury and decreased the effect of melatonin on oxidative stress, suggesting that melatonin inhibits H_2_O_2_-induced oxidative injury through upregulating the expression of NEAT1 [45]. Yan Q et al. showed that NEAT1 can promote oxidative damage during calcium oxalate (CaOx) crystal deposition by sponging miR-130a-3p [51]. Guo L et al. determined that the knockdown of NEAT1 inhibits angiogenesis mediated by exosomes derived from human umbilical vascular endothelial cells (HUVECs) stimulated by H_2_O_2_-induced oxidative stress [43,44]. Similarly, it has been shown that the knockdown of NEAT1 significantly decreases H_2_O_2_-induced oxidative stress in human cells [42,52]. Oxidative stress conditions induce TP53, which promotes the expression of NEAT1. Upregulated NEAT1 then suppresses H_2_O_2_-induced apoptosis through sponging miR-18a-5p [53]. Zou G et al. demonstrated that the knockdown of NEAT1 reduces apoptosis by inhibiting ROS production in high-glucose-induced cardiomyocytes [54]. Therefore, these studies suggest that NEAT1 is a critical player in the cellular response to oxidative stress. Oxidative stress can induce the expression of NEAT1, which is then critical for oxidative stress-induced damage and stress-related diseases.

### 4.3. NEAT1 Regulates Hypoxia

Hypoxia occurs when there is a deficiency of oxygen in cells. Cells detect low oxygen levels through hypoxia-inducible factors (HIFs). Under normal oxygen conditions, HIFs are degraded, but in hypoxia, they stabilize and activate genes involved in processes like angiogenesis, erythropoiesis, and metabolism [55]. Hypoxia can lead to the accumulation of ROS, resulting in oxidative stress.

Hypoxia in tumors can induce the transcription of NEAT1 and paraspeckle formation in a HIF-2α-dependent manner, leading to cancer cell survival [56]. NEAT1 can also be transcriptionally upregulated by hypoxia in non-cancer cells [57,58]. Furthermore, upregulated NEAT1 plays a critical role in hypoxia-induced apoptosis in normal cells, which results in hypoxia-related diseases. Myocardial ischemia–reperfusion (I/R) injury occurs when the blood supply returns to the heart tissue after a period of ischemia (a lack of oxygen). This process, while necessary to restore oxygen and nutrients, paradoxically causes additional damage to the heart tissue. During ischemia, HIFs are stabilized and activate genes that help cells adapt to low oxygen conditions. Upon reperfusion, the sudden influx of oxygen can lead to oxidative stress, which exacerbates tissue damage [59]. NEAT1 is significantly upregulated in myocardial IR injury mouse models. Additionally, NEAT1 is increased by hypoxia/reoxygenation in cells. Hypoxia-induced cardiomyocyte apoptosis can be reversed by the inhibition of NEAT1 [60]. Furthermore, the knockdown of NEAT1 protects against IR injury by decreasing cardiomyocyte apoptosis through the mitogen-activated protein kinase (MAPK) pathway. NEAT1 regulates hypoxia-induced apoptosis through other methods as well. For instance, NEAT1 induces hypoxia-induced apoptosis and autophagy through sponging miR-27a-3p, miR-181b, miR-29a, and miR-488-3p [61,62,63,64]. It is also shown that NEAT1 induces hypoxia/reoxygenation-induced cardiomyocyte injury through sponging miR-520a, miR-129-5p, miR-378a-3p, miR-339-5p, miR-22-3p, and miR-204 [65,66,67,68,69,70]. NEAT1 also regulates hypoxia-induced cardiomyocyte injury through promoting pri-miRNA processing [71]. Interestingly, NEAT1 is involved in regulating internal ribosome entry sites (IRESs) driving translation initiation in response to hypoxia in ischemic diseases [72]. Godet AC et al. determined that NEAT1 induces paraspeckle formation to serve as a platform to recruit IRES-containing mRNAs and IRESome, a multiple partner ribonucleic complex allowing ribosome recruitment onto mRNA, in response to hypoxia. High glucose or intermittent hypoxia can result in stress-induced damage in both cell and animal models of type 2 diabetes mellitus complicated with obstructive sleep apnea (T2DM-OSA) [73]. NEAT1 is induced by high glucose or intermittent hypoxia. The knockdown of NEAT1 can protect cells from high glucose or intermittent hypoxia-induced damage. Therefore, these studies suggest that NEAT1 plays a significant role in regulating cellular responses to hypoxia, which can lead to hypoxia-stress-induced diseases.

## 5. NEAT1 in Immune Cell Function

### 5.1. NEAT1 Regulates Innate Immune Cell

Innate immunity is the first line of defense against invading pathogens, such as viruses and microbes. Studies show that NEAT1 can be upregulated in immune cells in response to viral infection and promotes innate immunity against viral attack through various mechanisms (Figure 3 and Figure 4). For instance, the Hantaan virus (HTNV), commonly found in Asia, is the leading cause of severe hemorrhagic fever with renal syndrome (HFRS), resulting in a high mortality rate. Macrophages are critical for the host innate immune system, serving as the first line of defense against HTNV infection. NEAT1 is upregulated in monocytes and macrophages in mice and humans during the early HTNV infection. Additionally, the expression of NEAT1 in patient monocytes is negatively correlated with the HTNV load and the disease’s progression [74]. In mouse bone marrow-derived macrophages co-cultured with HTNV, the knockdown of Neat1 inhibits the activation of inflammatory macrophages, thereby facilitating HTNV infection. Conversely, the overexpression of the long isoform of Neat1, Neat1_2, significantly represses viral replication in the same co-culture experiment. A mechanistic study demonstrated that Neat1_2 may upregulate the expression of sterol regulatory element-binding protein 2 (Srebp2). Subsequently, Neat1_2 interacts with Srebp2 to stimulate the activation of inflammatory macrophages, thereby inhibiting HTNV infection [74]. Another study showed that NEAT1 is significantly upregulated in HUVECs infected with HTNV [75]. Again, they found that NEAT1_2, but not NEAT1_1, is responsible for defending HTNV infection. The depletion of NEAT1_2 in HUVEC dramatically promoted HTNV nucleocapsid protein (NP) production. The transient ectopic expression of NEAT1_2 repressed HTNV NP production, thereby efficiently suppressing HTNV replication. NEAT1 relocates splicing factor proline/glutamine-rich (SFPQ) to paraspeckles to derepress its transcriptional inhibitory effect on retinoic acid-inducible gene I (RIG-I) and dead/h-box helicase 60 (DDX60), thereby modulating the innate immune response against HTNV infection. Interestingly, the mechanism by which NEAT1 can activate transcription by relocating SFPQ has been demonstrated in an earlier study by Imamura K et al. [25]. In this study, they found that influenza virus infection, herpes simplex virus infection, or Toll-like receptor 3-p38 pathway-triggered poly I:C stimulation can induce the expression of NEAT1 in cultured stable cell lines. They further determined that upregulated NEAT1_2, but not NEAT1_1, relocated SFPQ to paraspeckles to derepress the expression of interleukin-8 (IL8) gene, resulting in an innate immune response. These studies suggest that the long isoform of NEAT1, NEAT1_2, is critical for defending against viral infection through paraspeckles in an innate immune response. It is interesting to note that interleukin-15 (IL15), but not IL8, can induce the transcription of NEAT1 by recruiting the signal transducer and activator of transcription 3 (STAT3) to its promoter in celiac disease (CD) patients [76]. Therefore, whether there is a regulatory feedback loop between NEAT1 and cytokines during viral infection warrants future investigations.

Morchikh M et al. identified another mechanism by which NEAT1 regulates the innate immune response in immune cells [77]. They showed that NEAT1 is required for the assembly of the HEXIM1-DNA-PK–paraspeckle components–ribonucleoprotein (HDP-RNP) complex, which includes hexamethylene bisacetamide inducible 1 (HEXIM1); the DNA-dependent protein kinase (DNA-PK) complex; SFPQ; and paraspeckle component 1 (PSPC1). The knockdown of HDP-RNP subunits had no impact on poly I:C stimulation but resulted in the loss of interferon (IFN)-stimulatory, DNA (ISD)-mediated IFNα; IFNβ; and MX dynamin-like GTPase 1 (MXA) induction. They further showed that the HDP-RNP complex is a positive regulator of DNA-mediated IFN response gene induction. Cyclic GMP-AMP synthase (cGAS); polyglutamine binding protein 1 (PQBP1); the stimulator of interferon genes (STING); interferon regulatory factor 3 (IRF3); and the mediators of innate immune activation all interact with the HDP-RNP complex. The innate immune response caused by ISD can lead to the remodeling of the HDP-RNP complex and the activation of both DNA-PKs and IRF3. Using Kaposi’s sarcoma-associated herpesvirus (KSHV) infection of HUVECs as a model, they demonstrated that the HDP-RNP complex is essential for the cGAS-dependent, KSHV-mediated activation of the innate immune response. It is noteworthy that this study did not show which isoform of NEAT1 is responsible for interacting with the HDP-RNP complex and the subsequent function.

The inflammasome plays a critical role in the innate immune response. Inflammasomes are multicomponent signaling platforms that control inflammatory responses. For instance, pathogen-associated molecular patterns (PAMPs) and damage-associated molecular patterns (DAMPs) activate one or more innate pattern-recognition receptors (PRRs). Once activated, the sensor proteins bind to and induce the oligomerization of an adaptor protein, namely an apoptosis-associated, speck-like protein containing a CARD (ASC), resulting in the formation of a single macromolecular aggregate, the ASC speck. Oligomerized ASC then recruits pro-caspase-1, which auto-processes to generate its activated form. Activated caspase-1 induces the proteolytic maturation of cytokine interleukin-1β (IL-1β) and interleukin-18 (IL-18), thereby inducing pyroptosis. Inflammasome activation is potent for defending against pathogens and damaged cells. However, dysregulated inflammasome activity can lead to autoimmune diseases, cancer, and neurodegenerative disorders.

In a mouse model of peritonitis and pneumonia, Zhang P et al. showed that Neat1 can interact with multiple proteins involved in inflammasomes to enhance their assembly, resulting in pro-caspase-1 processing [78]. For instance, the knockdown of Neat1 in murine, immortalized, bone-marrow-derived macrophages (iBMDMs) resulted in a significant reduction in the oligomerization of ASC, indicating the impaired assembly of the inflammasome. Conversely, the ectopic expression of Neat1 promoted the formation of the NLR family pyrin domain containing 3 (NLRP3) inflammasome, causing enhanced caspase-1 activation and IL-1β maturation. Mechanistically, Neat1 can enhance the activation of NLR family CARD domain containing 4 (NLRC4) and absent in melanoma 2 (AIM2) inflammasomes in an interleukin-6 (IL-6)-independent manner. Once stimulated with inflammasome-activating signals, Neat1 translocases from nuclear paraspeckles to the cytoplasm to modulate inflammasome activation by stabilizing mature caspase-1, promoting IL-1β production and pyroptosis. Lin J et al. showed that nickel–cobalt alloy magnetic nanocrystals (NiCo NCs) can inhibit the activation of NLRP3, NLRC4n and AIM2 inflammasomes [79]. Additionally, NiCo NCs inhibited neutrophil recruitment in a mouse model of acute peritonitis and reduced symptoms in another mouse model of colitis. Furthermore, NiCo NCs can reduce the expression of NEAT1, although the study did not show the functional role of NEAT1 in the NiCo NCs-mediated inhibition of inflammasomes. Therefore, these results suggest that NEAT1 upregulates inflammasomes to induce the innate immune response. It is also important to note that these studies did not specify which isoform of NEAT1 is responsible for the regulation of the inflammasomes.

Other immune-related diseases involved in the innate immune response are also related to the dysregulated expression of NEAT1. Fibrosis is a life-threatening disorder associated with tissue dysfunction caused by the excessive deposition of an extracellular matrix. Fibrosis is irreversible, and there is no effective treatment for it. Crosstalk between immune cells and other cells may play a critical role in the development of fibrosis. The deletion of RNA-binding motif protein 7 (Rbm7) in nonhematopoietic cells significantly repressed fibrosis in mice. NEAT1 interacts with RBM7. Furthermore, Rbm7 promotes the development of fibrosis by regulating cell death through inducing the dissociation of accumulated Neat1 paraspeckles, thereby preventing the repair of irreversibly damaged cells in a mouse model. Additionally, the in vivo delivery of Neat1 siRNA significantly induces fibrosis in a mouse model of Rbm7 knockout. Therefore, this study suggests a functional role of Neat1 in controlling fibrosis [80]. Though the study has not specified which isoform of Neat1 is critical for regulating fibrosis, it may indicate that NEAT1_2 plays a more important role, as the paraspeckle is involved in the function.

Several studies have indicated that NEAT1 may play a role in certain immune-related diseases through regulating the innate immune response, but no careful mechanistic study has been involved. For instance, Neat1 expression is significantly upregulated in the serum of mice and humans with inflammatory bowel disease (IBD) [81]. In this study, it was shown that NEAT1 can interact with tumor necrosis factor superfamily member 18 (TNFRSF18) mRNA and stabilize the mRNA. Subsequently, the upregulation of TNFRSF18 by NEAT1 activated the NF-kB signaling pathway, resulting in inflammation. Another study demonstrated that the knockdown of Neat1 reduced the intestinal epithelial barrier and promoted macrophage M1 transformation to M2, thereby reducing the inflammatory reaction [82]. Ulcerative colitis (UC) is caused by the dysfunction of intestinal epithelial cell (IEC)-mediated intestinal epithelial barrier damage. Ni S et al. identified the upregulation of NEAT1 in IECs from UC patients. Furthermore, NEAT1 can promote glucose metabolism by sponging miR-410-3p, thereby resulting in LPS-induced IEC dysfunction [83]. NEAT1 is significantly downregulated in the PMBCs of early-onset myocardial infarction (MI) patients, mainly in the cluster of differentiation-14-positive (CD14^+^) monocytes [84]. Hepatitis B virus (HBV) infection remains a major public health issue, as many people die from HBV-associated liver diseases. Zeng Y et al. found that the expression of NEAT1 is significantly downregulated in the peripheral blood of patients with chronic HBV infection compared to healthy donors. They suggest that chronic HBV infection (CHB) can repress the innate immune response by downregulating NEAT1 [85]. Neat1 knockout macrophages failed to express a large set of proinflammatory cytokines, chemokines, and antimicrobial mediators, leading to a failure to control the replication of salmonella enterica serovar Typhimurium or vesicular stomatitis virus [86]. In a mouse model of myocarditis, transfusion with Neat1-low dendritic cells (DCs) reduced the inflammatory response, while in a heart transplantation model, transfusion with Neat1-low DCs induced immune tolerance. Additional experiments indicate that the upregulation of Neat1 can lead to the generation of tolerogenic DCs [87].

Overall, these studies suggest that NEAT1 plays a significant role in regulating the innate immune response. Most studies have shown that the long isoform of NEAT1 plays a significant role in regulating the innate immune response, while many studies did not specify which isoform is critical for defending against disease. Specifying which isoform of NEAT1 is responsible for the regulation will not only help determine the underlying mechanisms but also help develop therapeutic strategies against viral infections and other pathogens that threaten humans.

### 5.2. NEAT1 Regulates Adaptive Immune Cell Function

Besides the innate immune response, NEAT1 may also regulate the adaptive immune response (Figure 3 and Figure 4). NEAT1 expression is significantly upregulated in PBMCs of HIV-1-infected patients. The expression of NEAT1 is significantly reduced in HIV-1-infected patients treated with highly active antiretroviral therapy, suggesting that NEAT1 may be involved in the immune response to HIV-1 infection [88,89,90]. Liu H et al. further found that NEAT1 is significantly decreased following the activation of PBMCs and CD4^+^ T cells in response to HIV-1 infection [91]. Furthermore, HIV-1 replication is enhanced in Jurkat cells with NEAT1 knockout compared to wild-type Jurkat cells. These results suggest a potential role of NEAT1 in regulating adoptive immune cells, particularly CD4^+^ T cells.

The occurrence and development of rheumatoid arthritis (RA) is mainly caused by Th17 cells being differentiated from CD4^+^ T cells. NEAT1 is significantly increased in the PBMCs of RA patients and is also significantly upregulated in Th17 cells that are differentiated from CD4^+^ T cells in vitro. The knockdown of NEAT1 inhibits CD4^+^ T cells from differentiating into Th17 cells. NEAT1 upregulates the expression of STAT3 protein, which is critical for Th17 cell differentiation. In a type II collagen-induced arthritis mouse model, the in vivo injection of NEAT1 siRNA decreased the number of Th17 cells, thereby reducing the arthritis severity [92]. Another example of NEAT1 regulating CD4^+^ T cell differentiation is coronary heart disease (CHD), the most common cardiovascular disease caused by atherosclerosis. The expression of NEAT1 is significantly higher in the plasma of CHD patients compared to normal patients [93]. Additionally, the expression of NEAT1 is positively correlated with tumor necrosis factor alpha (TNFα), IL-1β, IL-6, and interleukin-17 (IL-17) in the plasma of CHD patients. Furthermore, the expression of NEAT1 is also positively correlated with the percentage of Th17 cells, but not Th1 or Th2 cells, in the plasma of CHD patients, suggesting that NEAT1 may positively regulate CD4^+^ T cells differentiating into Th17 cells. Autoimmune uveitis (AU) is an immune-related intraocular inflammatory disorder affecting millions of people worldwide. It mainly occurs in young adults and is characterized by inflammatory cell infiltration and ocular damaging, resulting in visual impairment and blindness [94]. Chen S et al. identified that NEAT1 is significantly upregulated in the PBMCs of AU patients [95]. In a mouse model of experimental autoimmune uveitis (EAU), they further showed that Neat1 expression is significantly increased in CD4^+^ T cells in mice with EAU compared to normal mice. Additionally, Neat1 expression is positively correlated with disease severity. Furthermore, Neat1 is significantly upregulated in Th17 cells compared with T helper 0 (Th0) and Th1 cells. They showed that Neat1 induced by interleukin-23 (IL-23)/STAT3 signaling may be critical for the progression of EAU in mice, supported by the evidence that the lentivirus-mediated knockdown of Neat1 in mice alleviated EAU compared to control lentivirus-treated mice. They further showed that Neat1 may promote EAU by upregulating the Th17 cell response. Therefore, these studies demonstrate that NEAT1 is critical for CD4^+^ T cells differentiating into Th17 cells, thereby regulating the adaptive immune response.

NEAT1 may also regulate CD4^+^ T cells differentiating into other subtypes, in addition to Th17 cells. Atherosclerosis and myocardial infarction are mainly caused by inflammation. NEAT1 is significantly downregulated in the PBMCs of post-MI patients [86]. It is observed that there are more Th17 cells and T regulatory (Treg) cells in the spleens of Neat1 knockout (Neat1^−/−^) mice compared to wild-type mice, suggesting a shift of CD4^+^ T cell balance towards Th17 cells and Treg cells. As the most abundant allergic disease in the pediatric population, allergic rhinitis (AR) is a hypersensitivity reaction induced by the immunoglobulin E (IgG)-mediated inflammatory response, resulting in nasal itching, congestion, rhinorrhea, and sneezing. It is reported that the expression of NEAT1 is significantly higher in the PBMCs of AR patients compared to normal patients [96]. Interestingly, the expression of NEAT1 is positively correlated with Th2 cells, but not Th1 cells, in AR patients. However, the correlation between NEAT1 and Th17 cells was not examined in the study. Similarly, the expression of NEAT1 is significantly upregulated in CD4^+^ T cells in the peripheral blood of children with asthma [97]. Additionally, the knockdown of NEAT1 significantly decreased Th2-related cytokines but had no impact on Th1 cells. Another study suggested that NEAT1 could also be critical for regulating the Th1/Th2 balance in the patients infected with SARS-CoV-2 (COVID-19) [98]. The overexpression of NEAT1 in CD4^+^ T cells promotes the levels of Th2-related cytokines interleukin-4 (IL-4), interleukin-5 (IL-5), and interleukin-13 (IL-13) [99]. In contrast, Th1-related cytokine IFN-γ production is reduced by the upregulation of NEAT1 in CD4^+^ T cells [99]. NEAT1 promotes signal transducer and activator of transcription 6 (STAT6) protein expression but has no impact on its mRNA level, as NEAT1 interacts with STAT6 protein to prevent its ubiquitination. Therefore, NEAT1 promotes Th2 cell activation through STAT6, supporting the observation that NEAT1 expression is induced in the PBMCs of systemic lupus erythematosus (SLE), which is primarily influenced by a Th2-mediated immune response [99]. In another study, Yan X et al. observed the significant upregulation of NEAT1 in the CD4^+^ T cells from the PBMCs of children with asthma. Mechanistic studies revealed that the knockdown of NEAT1 significantly impaired the production of Th2-related cytokines, thereby inhibiting Th2 differentiation [97]. A shift of the CD4^+^ T cell balance towards T helper cell proliferation was observed both in circulating blood and spleens in Neat1^−/−^ mice compared to Neat1 wild-type mice [84]. Therefore, these studies suggest that NEAT1 is critical for regulating the balance between Th1/Th2 in these patients.

In addition to regulating the differentiation of T cells, NEAT1 can also promote the survival of T cells. For instance, sepsis is a life-threatening disease initiated by cytokine-mediated hyperinflammation. NEAT1 expression is significantly upregulated in the PBMCs of sepsis patients. In a mouse model of sepsis, the knockdown of Neat1 accelerates T lymphocyte viability and reduces apoptosis [100]. Mechanistic studies showed that Neat1 can induce the expression of mast cell-expressed membrane protein 1 (MCEMP1) by sponging microRNA-125 (miR-125) to promote the serum levels of inflammatory factors, T lymphocyte activity, and apoptosis, thereby increasing immunity in septic mice.

The correlation between NEAT1 expression and B cells is less known. Qiu Y et al. conducted the single-cell RNA sequencing of cluster of differentiation-19-positive (CD19^+^) cells in patients with Waldenström’s macroglobulinemia (WM), a rare lymphoproliferative disorder [101]. They found that NEAT1 expression is significantly upregulated in mature B cells of WM patients. Chattopadhyay P et al. found that the expression of NEAT1 is significantly upregulated in the PBMCs of patients infected with COVID-19 [102]. Additionally, they identified that NEAT1 was significantly upregulated in naïve B cells, naïve CD4^+^ cells, and naïve cluster of differentiation-8-positive (CD8^+^) T cells in the infected individuals.

Overall, these studies have demonstrated the critical role of NEAT1 in regulating the adaptive immune response, particularly in the differentiation of CD4^+^ T cells. However, none of these studies have specified which isoform of NEAT1 is responsible for regulating this differentiation process. Many studies have used Neat1^−/−^ mouse, which shows the significant downregulation of both isoforms of Neat1. The authors who created this knockout mouse have recently generated another mouse that solely expresses the long isoform by using CRISPR/Cas9-mediated deletion of the PAS domain, which is essential for generating the short isoform [27]. This mutant mouse can be used to determine whether the deletion of the short isoform of Neat1 impacts immune cell regulation.

## 6. NEAT1 as a Biomarker and Therapeutic Target in Stress- and Immune-Related Diseases

### 6.1. NEAT1 as a Biomarker and Therapeutic Target in Stress-Related Diseases

Several studies have shown that NEAT1 is upregulated in stress-related diseases, the pathogenesis and treatment options for which remain unknown (Table 1). For instance, Chen R et al. determined that NEAT1 was significantly induced in a mouse model of 1-methyl-4-phenyl-1,2,3,6-tetrahydropyridine (MPTP)-induced Parkinson’s disease, suggesting that NEAT1 may serve as a biomarker for this disease [103]. Senousy M et al. explored non-invasive biomarkers for preeclampsia patients [104]. They found that NEAT1 expression in serum is significantly correlated with ultrasound data diagnosing preeclampsia patients with early onset, further suggesting it as a novel serum biomarker for these patients. Similarly, Dou X et al. suggested that NEAT1 can serve as a diagnostic biomarker and therapeutic target for patients with pulmonary arterial hypotension, as it is highly expressed in hypoxia-treated pulmonary atrial smooth muscle cells and the serum of the patients [105]. Additionally, it was shown that the silencing of NEAT1 can reduce other stress-related diseases in preclinical studies. Yu Q et al. showed that the knockdown of NEAT1 decreased myocardium ischemia both in in vitro and in vivo animal studies [106]. NEAT1 repression improved autism-related behaviors [48]. The knockdown of NEAT1 significantly inhibited the angiogenic capacity of endothelial progenitor cells, which is required for the survival of skin flaps during transplantation in reconstructive surgery [44]. The knockdown of NEAT1 reduced apoptosis and necrosis in mouse cardiomyocytes or hypoxia/reoxygenation-induced injury in human umbilical vein endothelial cells, suggesting that NEAT1 can be a potential target in cardio protection in myocardial infraction patients [70,71,107]. Conversely, Simchovitz A suggested that NEAT1 may serve a protective role in Parkinson’s disease, via the depletion of NEAT1-induced cell death after paraquat-induced oxidative stress [108]. However, it is important to note that all these summarized works remain either in the correlation research stage or at the early stages of preclinical studies (Table 1). Therefore, NEAT1 serving as a biomarker or a therapeutic target for those diseases warrants further investigation and more extensive preclinical and clinical studies.

### 6.2. NEAT1 as a Biomarker and Therapeutic Target in Immune-Related Diseases

As mentioned above, NEAT1 is dysregulated in the PBMCs of patients with multiple immune-related diseases, suggesting that NEAT1 can serve as a biomarker for assisting disease management and prognoses (Table 1). Furthermore, some studies demonstrate that NEAT1 may be a therapeutic target in immune-related diseases (Table 1).

Li P et al. explored the correlation of NEAT1 with disease severity and recurrence in patients with acute ischemic stroke (AIS) [114]. NEAT1 is significantly upregulated in PBMCs of patients with AIS compared to healthy donors. Additionally, NEAT1 expression positively correlates with the National Institutes of Health Stroke Scale (NIHSS) score. Furthermore, NEAT1 expression is also positively correlated with increased recurrence and death risk and negatively correlated with recurrence-free survival. Therefore, NEAT1 can serve as a biomarker for AIS. NEAT1 expression is significantly upregulated in sepsis patient samples. There is a significant positive correlation between NEAT1 and the severity of sepsis, indicating that NEAT1 may serve as a potential biomarker for sepsis [119,120,121]. The expression of NEAT1 is significantly high in a subset of PBMCs driven by sepsis. Huang Q et al. also showed that NEAT1 expression is significantly upregulated in the plasma and serum of sepsis patients. Its expression is significantly associated with a higher sepsis risk [120]. Additionally, the higher expression of NEAT1 is associated with a good prognosis in sepsis patients. The early diagnosis of Behçet’s disease (BD), a chronic autoimmune disease, is critical to avoid serious and fatal complications. Mohammed SR et al. and Mohammed A et al. found that NEAT1 expression in serum is significantly correlated with the severity of the disease, suggesting that it can be a biomarker for BD [122,123]. Pediatric immune thrombocytopenic purpura (ITP) is an autoimmune disease. Hamdy SM et al. found that NEAT1 expression is significantly upregulated in the sera of children with ITP [124]. Furthermore, they also found that there is a significant upregulation of NEAT1 in non-chronic compared to chronic ITP patients. The platelets under apoptosis that are presented to T lymphocytes with the aid of dendritic cells are important in the pathogenesis of ITP. There is a significant negative correlation between NEAT1 and platelet counts before treatment. Therefore, NEAT1 expression in serum can serve as a potential biomarker for differentiating childhood ITP patients from healthy individuals, in addition to differentiating non-chronic from chronic ITP patients. NEAT1 expression is significantly upregulated in the PBMCs of patients with acute/chronic inflammatory demyelinating polyradiculoneuropathies (AIDP/CIDP) [125]. As the first evidence showing the expression of NEAT1 positively correlates with this disease, further studies are warranted to elucidate the molecular mechanisms of NEAT1 in AIDP/CIDP. It also suggests that NEAT1 can serve as a biomarker for AIDP/CIDP. Atypical hemolytic uremic syndrome (aHUS) is a rare and fatal thrombotic microangiopathy. However, the biomarker for diagnosing this disease is still unknown. Therefore, Chen I et al. conducted single-cell sequencing on the PBMCs of aHUS patients and normal controls. They observed the significant upregulation of NEAT1 in the PBMCs of unstable aHUS patients, providing valuable insights into potential biomarkers for this disease [126]. NEAT1 can potentially be a biomarker for autoimmune disease but also a potential biomarker in subgrouping the disease. For instance, Hamdy SM et al. detected that NEAT1 is significantly upregulated in the sera of pediatric ITP patients. Additionally, NEAT1 is further significantly upregulated in the sera of non-chronic in comparison to chronic ITP patients [124]. The expression of NEAT1 is found to be significantly upregulated in the plasma samples of AD patients, suggesting that it can be a promising diagnosis biomarker for this disease [127]. Ballonová L et al. showed that NEAT1 is significantly upregulated in monocytes of a group of Hereditary angioedema (HAE) patients. Frailty, an intermediate status of the human aging process, has been correlated with decompensated homeostasis and mortality [128]. Luo OJ et al. conducted a single-cell analysis in PBMCs from healthy young adults and frail, old adults to investigate the immune phenotype of frailty, which is still poorly understood [129]. Interestingly, they identified a specific subset of monocyte exclusively identified in the frailty group. Furthermore, this subset of monocytes exclusively expresses a high level of NEAT1. It is suggested that NEAT1 can also serve as a marker for a subset of monocytes [130]. Multiple studies revealed that NEAT1 expression is highly expressed in PBMCs and the saliva of mild and severe patients infected with COVID-19 [110,111,112,113]. Additionally, the expression of NEAT1 positively correlates with the severity of the virus infection risk. Overall, these studies suggest that NEAT1 can serve as a biomarker for immune-related diseases.

Upregulated NEAT1 regulates immune response in immune-related diseases, indicating itself as a potential therapeutic target. For instance, several studies explored NEAT1 as a potential therapeutic target in IBD [82,131]. NEAT1 expression is significantly higher in the serum of IBD mice compared to normal controls. Furthermore, they found that the knockdown of NEAT1 significantly suppresses the dextran sulfate sodium (DSS)-induced permeability increase in colon tissues from IBD mice, possibly through promoting the transformation of macrophage M1 to M2 and suppressing the inflammatory reaction. Ni X et al. examined NEAT1 as a therapeutic target in cerebral ischaemia/reperfusion (I/R) injury [109]. The knockdown of NEAT1 inhibited apoptosis in N2a cells caused by cerebral I/R injury. Additionally, the knockdown of NEAT1 reduced the AKT/STAT3 pathway in BV-2 cells caused by deprivation/reoxygenation (OGD/R) injury. Furthermore, the knockdown of NEAT1 suppressed M1 microglial polarization in OGD/R-exposed microglial cells. The in vivo delivery of a lentivirus to express a NEAT1 siRNA to target Th17 cells relieved RA in a type II collagen-induced mouse model of rheumatoid arthritis [92,132]. Ye L et al. observed a significant upregulation of NEAT1 in peripheral T cells, including both the CD4^+^ and CD8^+^ T cells, of patients with primary Sjogren’s syndrome (pSS), a systemic autoimmune disease [133]. The inhibition of NEAT1 by antisense increased the expression of the C-X-C motif chemokine ligand 8 (CXCL8) and TNF-α in PMA/ionomycin-stimulated Jurkat cells. The study highlights the potential of NEAT1 as a therapeutic target of pSS. The intraventricular injection of an antisense oligonucleotide to knock down NEAT1 significantly reduced brain damage by reducing activated microglia and proinflammatory cytokines in a mouse model of ischemic stroke, supporting NEAT1 as a potential therapeutic target for ischemic stroke treatment [115,116]. Overall, these studies indicate that NEAT1 has the potential to be a therapeutic target for treating multiple immune-related diseases.

### 6.3. Current Limitations of the Application of NEAT1 as a Therapeutic Target

NEAT1, among other lncRNAs, holds promise as a therapeutic target in both cancers and non-cancerous diseases. However, several limitations needed to be addressed to fully realize its potential. Notably, none of the existing studies have focused on targeting the specific isoforms of NEAT1 to modulate these diseases. The antisense oligonucleotide used in the studies targets both isoforms. Therefore, extensive research is needed to determine which isoform should be targeted for therapeutic purposes in future studies, given that two isoforms may have differential functions in the cells. Additionally, all current studies are limited by the use of either in vitro cell lines or in vivo studies at early stages. More in vivo work is necessary to further demonstrate NEAT1 as a therapeutic target for relevant diseases. For instance, developing specific delivery vehicles that can accurately target NEAT1-modulating reagents to the desired cells or tissues is complex and requires extensive research. Ensuring specificity and minimizing off-target effects is critical for the safety and efficacy of NEAT1-based treatments. Overall, lncRNA-based treatments are rare in clinical trials [117,118]. Therefore, more clinical research is warranted to validate the therapeutic potential of lncRNAs.

## 7. Conclusions

NEAT1 regulates the response to stress and plays an important role in stress-related diseases. Additionally, NEAT1 regulates both innate and adaptive immune systems in immune-related diseases. The upregulation of NEAT1 by virus infection and its positive correlation with the severity of immune-related diseases suggest that NEAT1 can serve as a biomarker for these conditions. The modulation of NEAT1 has been shown to suppress diseases in vivo, further indicating the NEAT1 can be a therapeutic target. Future research on NEAT1 in immune-related disease should focus on dissecting the precise molecular mechanisms by which NEAT1 regulates stress and immune responses and contributes to disease pathogenesis. Investigating the interaction between NEAT1 and different signaling pathways, as well as exploring NEAT1 as a potential biomarker for early detection and therapeutic targeting, could open new avenues for treatment. Additional studies are required to determine which isoform of NEAT1 should be targeted for therapeutic purposes. Several engineering methods have been introduced to modulate the expression of each isoform of NEAT1 in cells [23] and in mice [27]. Those tools should be carefully chosen to determine the role of each NEAT1 isoform in the immune-related diseases and identify the appropriate target. Additionally, advanced techniques, like clustered regularly interspaced short palindromic repeats/cas13d (CRISPR/Cas13d), could be potentially employed to modulate the expression of each NEAT1 isoform and assess its effects in preclinical models [134,135]. Overall, a deeper understanding of NEAT1’s functions and regulatory networks may lead to novel therapeutic strategies for managing stress- and immune-related diseases.

## Figures and Tables

**Figure 1 ijms-26-04413-f001:**
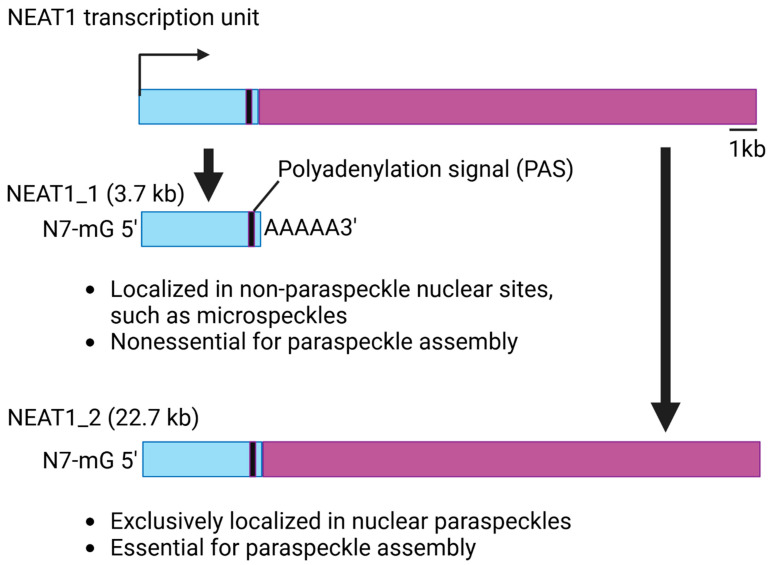
The structural feature and cellular functions of NEAT1.

**Figure 2 ijms-26-04413-f002:**
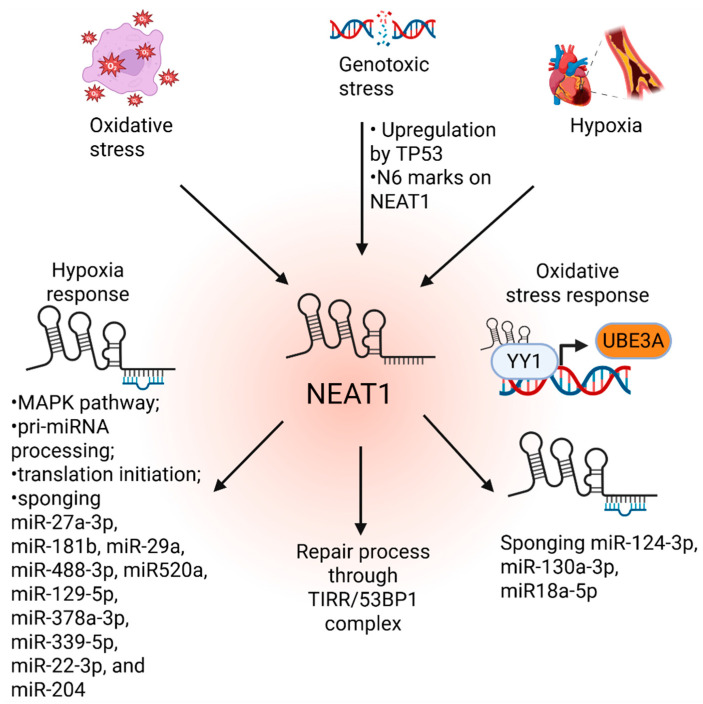
The role of NEAT1 in the stress response.

**Figure 3 ijms-26-04413-f003:**
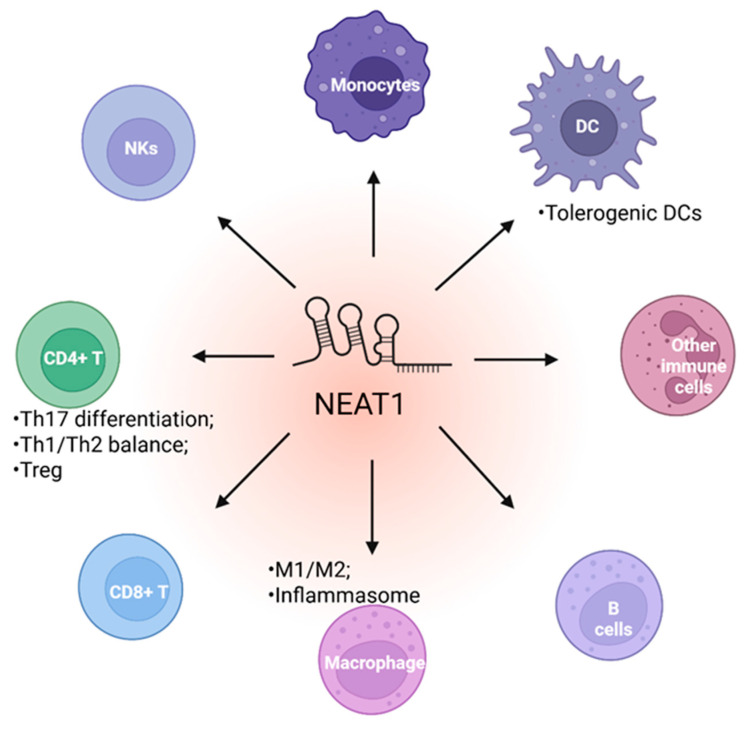
The role of NEAT1 in various immune cells.

**Figure 4 ijms-26-04413-f004:**
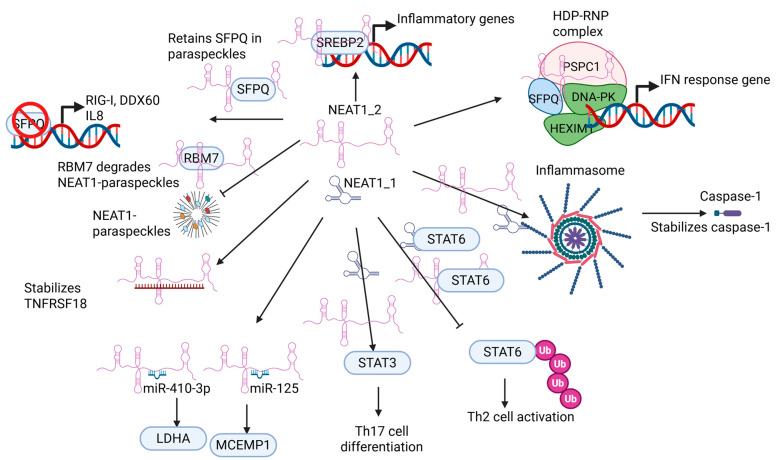
NEAT1 regulates immune cells by interacting with various proteins and miRNAs. “Ub” stands for ubiquitination.

**Table 1 ijms-26-04413-t001:** The overview of NEAT1 in different stress- and immune-related diseases.

	DISEASE TYPE	NEAT1 Expression	Mechanisms	References (Ref.)
Stress-related	Neurodegenerative diseases (Huntington’s Disease, Parkinson’s Disease, Huntington’s disease, amyotrophic lateral sclerosis)	Upregulated	Induced by genotoxic stress, regulating genome integrity through TIRR/53BP1 complex	[32,33,34,35,36,37,38]
Stress-related	Diabetic retinopathy	Upregulated	Induced in mitochondria by high glucose	[39]
Stress-related	Autism spectrum disorder	Upregulated	NEAT1 promotes apoptosis and oxidative stress through UBE3A	[48]
Stress-related	Temporomandibular joint disorders	Upregulated	NEAT1 promotes ROS in mitochondria	[49]
Stress-related	Intervertebral disc degeneration	Upregulated	NEAT1 induces oxidative stress-induced apoptosis through sponging miR-124-3p	[50]
Stress-related	Myocardial ischemia-reperfusion injury	Upregulated	NEAT1 induces apoptosis through MAPK pathway, sponging miRNAs, promoting pri-miRNA processing, and promoting translation initiation	[54,59,60,61,62,63,64,65,66,67,68,69,70,71,72,106,109]
Stress-related	Type 2 diabetes	Upregulated	NEAT1 regulates hypoxia-induced damage	[73]
Immune-related	Influenza	Upregulated	NEAT1 induces transcription of IL8	[25]
Immune-related	Hantaan virus	Upregulated	NEAT1 activates inflammatory macrophages through Srebp2	[74,75]
Immune-related	Celiac disease	Downregulated	IL15 induces transcription of NEAT1	[76]
Immune-related	Kaposi’s sarcoma-associated herpesvirus	N/A	NEAT1 activates innate immune response through HDP-RNP complex	[77]
Immune-related	Peritonitis and pneumonia	N/A	NEAT1 activates NLRP3, NLRC4, and AIM2 inflammasomes	[78]
Immune-related	Fibrosis	N/A	NEAT1 represses fibrosis through interacting with Rbm7	[80]
Immune-related	Inflammatory bowel disease	N/A	Knockdown of NEAT1 promotes macrophage M2	[81,82,83,107]
Immune-related	Atherosclerosis and myocardial infarction	Downregulated	NEAT1 regulate generation of tolerogenic DCs and CD4^+^ T cell balance	[84]
Immune-related	Hepatitis B virus	Downregulated	N/A	[85]
Immune-related	Human immunodeficiency virus 1	Upregulated	N/A	[88,89,90]
Immune-related	Rheumatoid arthritis	Upregulated	NEAT1 promotes CD4^+^ T cells differentiating into Th17 cells	[92]
Immune-related	Coronary heart disease	Upregulated	NEAT1 promotes CD4^+^ T cells differentiating into Th17 cells	[95]
Immune-related	Autoimmune uveitis	Upregulated	NEAT1 promotes CD4^+^ T cells differentiating into Th17 cells	[94,95]
Immune-related	Allergic rhinitis	Upregulated	NEAT1 regulates Th1/Th2 balance	[96]
Immune-related	Asthma	Upregulated	NEAT1 promotes Th2 cell activation	[97,100]
Immune-related	SARS-CoV-2	Upregulated	N/A	[98,102,110,111,112,113]
Immune-related	Systemic lupus erythematosus	Upregulated	NEAT1 promotes Th2 cell activation through STAT6	[99]
Immune-related	Waldenström’s macroglobulinemia	Upregulated	N/A	[101]
Immune-related	Parkinson’s disease	Upregulated	N/A	[102,108]
Immune-related	Preeclampsia	Upregulated	N/A	[104]
Immune-related	Pulmonary arterial hypotension	Upregulated	N/A	[105]
Immune-related	Acute ischemic stroke	Upregulated	N/A	[114,115,116,117,118]
Immune-related	Sepsis	Upregulated	NEAT1 induces MCEMP1 by sponging miR-125	[119,120,121]
Immune-related	Behçet’s disease	Upregulated	N/A	[122,123]
Immune-related	Immune thrombocytopenic purpura	Upregulated	N/A	[124]
Immune-related	Acute/chronic inflammatory demyelinating polyradiculoneuropathies	Upregulated	N/A	[125]
Immune-related	Atypical hemolytic uremic syndrome	Upregulated	N/A	[126]
Immune-related	Alzheimer’s disease	Upregulated	N/A	[127]
Immune-related	Hereditary angioedema	Upregulated	N/A	[128]
Immune-related	Sjögren’s syndrome	Upregulated	N/A	[129]

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
