# Peer review of "Emerging Role of Long, Non-Coding RNA Nuclear-Enriched Abundant Transcript 1 in Stress- and Immune-Related Diseases"

_ijms, 2025, doi:10.3390/ijms26094413_

Round 1
Reviewer 1 Report (New Reviewer)
Comments and Suggestions for Authors
Please see the attachment.

There are many grammatical mistakes and sentences without proper meaning. The authors need to work on that.
Author Response
The manuscript entitled “Emerging role of non-coding RNA NEAT1 in stress and immunerelated
diseases” by Liu et al. is a comprehensive review focused on our current
understanding of the role of the long non-coding RNA (lncRNA) NEAT1 in diseases
associated with stress and immunological response. The authors presented an overview of
lncRNAs and introduced a particular lncRNA – NEAT1 which has gained a lot of attention in
last one decade because of its multifaceted role in human health and disease. This was
followed by a detailed discussion on the involvement of NEAT1 in di?erent stress response
mechanisms including genotoxic stress, hypoxia etc. and immune cell function. The
authors also explored the function of NEAT1 as a biomarker as well as therapeutic targets
in stress and immune-related diseases. Overall, I enjoyed reading the article and
appreciate the authors’ e?ort to write a review on this important lncRNA. My specific
comments are summarized below.
JH: Thank you for enjoying reading this manuscript.
Major comments:
- Lellahi et al. convincingly showed that NEAT1 expression and nuclear paraspeckles
are elevated by heat shock in heat shock factor 1 (HSF1)-dependent manner (PMID:
30305397). The authors should include this article (and if there are other articles on
heat stress and NEAT1) as without this, the discussion on the involvement of NEAT1
in stress response remains incomplete.
JH: Thank you for your suggestion. Yes, we agree that Lellahi et al. convincingly showed the regulation of NEAT1 by HSF1. We did not include this article along with others on heat stress and NEAT1 because those studies examined NEAT1 in stress response using cancer cell lines, such as MCF7 and Hela cancer cell lines in PMID: 30305397. This is out of the scope of this manuscript, which focuses on the role of NEAT1 in non-cancerous diseases.
- The authors briefly mentioned the role of NEAT1 in Alzheimer’s Disease in the text
and the table. There are very important articles/reviews on role of NEAT1 on other
neurodegenerative diseases like Huntington’s Disease, Parkinson’s Disease etc.
(e.g., PMID: 30321100, PMID: 30533572, PMID: 33325399). These look like an
obvious omission unless there is a clear rationale behind mentioning Alzheimer’s
Disease but excluding other neurodegenerative diseases.
JH: Thank you for your suggestion. Yes, Li K et al (PMID: 36738893) and others have extensively reviewed NEAT1 in neurodegenerative diseases. We reviewed the role of NEAT1 in stress- and immune-related diseases. We only included articles indicating that NEAT1 plays a role in stress response and immune regulation, which are critical for neurodegenerative diseases. However, we believe it will be a good idea to include these references on NEAT1 in other neurodegenerative diseases, given that they are often affected by stress. Therefore, we have included these references in the revision.
Minor points:
- Ref. 115 is in the wrong place – in the middle of the discussion on immune cells.
JH: Thank you for pointing out the wrong citation. We meant to cite the PMID: 36947499, which showed upregulation of NEAT1 in the plasma samples of patients with Alzheimer’s disease. We have updated this citation in the revision.
- The full names of PBMCs, HIV etc. should be when these terms are first introduced
and not randomly somewhere in the manuscript.
JH: Thank you for pointing it out. We have ensured that these abbreviations are introduced when they first appear in the revision.
- There are many typos and incomplete sentences throughout the manuscript that
should be corrected.
JH: Thank you for pointing that out. We have corrected those typos and incomplete sentences. Additionally, we requested a native English speaker to further improve our manuscript.
- The quality of English needs to be improved. The authors might consider getting the
manuscript checked by a native English speaker.
JH: Thank you for your recommendation. As we mentioned above, we have requested a native English speaker to improve our manuscript.

Reviewer 2 Report (New Reviewer)
Comments and Suggestions for Authors
The manuscript by Liu et al., entitled Emerging role of long non-coding RNA NEAT1 in stress and immune-related diseases, is well written and easy to follow.
To enrich the manuscript, I suggest adding a chapter describing the current limitations of the application of NEAT1 as a therapeutic target. I suggest adding a graphical abstract that would encourage the readers.
After these changes, I recommend the manuscript for publication.
Author Response
The manuscript by Liu et al., entitled Emerging role of long non-coding RNA NEAT1 in stress and immune-related diseases, is well written and easy to follow.
To enrich the manuscript, I suggest adding a chapter describing the current limitations of the application of NEAT1 as a therapeutic target. I suggest adding a graphical abstract that would encourage the readers.
After these changes, I recommend the manuscript for publication.
JH: Thank you for your kind comments and suggestions. We have added a specific chapter describing the current limitations of the application of NEAT1 as a therapeutic target in the end of the section “NEAT1 as a biomarker and therapeutic target in immune-related diseases”:
“Current limitations of the application of NEAT1 as a therapeutic target
NEAT1, among other lncRNAs, holds promise as a therapeutic target in both cancers and non-cancerous diseases. However, several limitations needed to be addressed to fully realize its potential. Notably, none of the existing studies have focused on targeting specific isoforms of NEAT1 to modulate these diseases. The antisense oligonucleotide used in the studies targets both isoforms. Therefore, extensive research is needed to determine which isoform should be targeted for therapeutic purposes in future studies. Additionally, all current studies are limited by the use of either in vitro cell lines or in vivo studies at early stages. More in vivo work is necessary to further demonstrate NEAT1 as a therapeutic target for relevant diseases. For instance, developing specific delivery vehicles that can accurately target NEAT1-modulating reagents to the desired cells or tissues is complex and requires extensive research. Ensuring specificity and minimizing off-target effects is critical for the safety and efficacy of NEAT1-based treatments. Overall, lncRNA-based treatments are rare in clinical trials 134,135. Therefore, more clinical research is warranted to validate the therapeutic potential of lncRNAs.”
We also added a graphical abstract overviewing the manuscript on NEAT1.

This manuscript is a resubmission of an earlier submission. The following is a list of the peer review reports and author responses from that submission.
Round 1
Reviewer 1 Report
Comments and Suggestions for Authors
The role of lncRNAs in regulating a wide range of cellular processes is becoming more and more evident, thus the idea of a review on the role of NEAT1 in immune-related diseases is interesting.
However, the submitted paper does not provide a comprehensive view of the subject, in particular it seems a collection of sentences that more or less summarize different papers without providing a critical view of the results obtained by other authors. In addition, in different part of the review, the authors are quoting the interaction of NEAT1 with several transcription factors/proteins, but it is extremely difficult to understand the interactions without a figure.
Lastly, the authors included diseases such as atherosclerosis but left out other diseases in which the immune component is more relevant, such as IBD (only one paper is cited) or celiac disease.
Comments on the Quality of English LanguageEnglish language could be improved
Author Response
Thank you for your valuable suggestions. I have incorporated your feedback into our revised manuscript: 1) We have included more studies on IBD (Ref: 13, 47-49, 74), and a study on celiac disease (Ref: 42); 2) We have provided some critical review of the results (such as Page 7 lines 156-160, Page 10 lines 209-211); 3) We have added a new figure (Figure 3) to overview NEAT1 interacting proteins and microRNAs in regulating immune cell functions.
Reviewer 2 Report
Comments and Suggestions for Authors
The manuscript presents literature data about the role of lncRNA NEAT1 in immune immune-related diseases.
Meanwhile, I have some suggestions to improve the manuscript:
In the abstract authors should precise the concrete goal of the review, for example, a new hypothesis for the role of the lncRNA NEAT1 or a new mechanism of action etc.
The Section “Introduction” must be short and present part of the literature data according to the main purpose of the article. The present form of the manuscript does not contain an introduction. All data presented in the “Introduction” consists of the main body of the manuscript.
Section “Conclusion” is too long and needs rewriting focusing on the new approach reaching through the manuscript.
Author Response
Thank you for your valuable suggestions to improve our manuscript. We have incorporated all your suggestions into the revised manuscript. 1) We have added one sentence to clarify the goal of the review; 2) We have added a paragraph to the “Introduction” section; 3) We also reduced the “Conclusion” section. All these changes are highlighted in the revised manuscript using track changes.